



# Nemo-Nordic 2.0: Operational marine forecast model for the Baltic Sea

Tuomas Kärnä[1], Patrik Ljungemyr[2], Saeed Falahat[2], Ida Ringgaard[3], Lars Axell[2], Vasily Korabel[3], Jens Murawski[3], Ilja Maljutenko[4], Anja Lindenthal[5], Simon Jandt-Scheelke[5], Svetlana Verjovkina[4], Ina Lorkowski[5], Priidik Lagemaa[4], Jun She[3], Laura Tuomi[1], Adam Nord[2], and Vibeke Huess[3]

[1]Finnish Meteorological Institute, Helsinki, Finland
[2]Swedish Meteorological and Hydrological Institute, Norrköping, Sweden
[3]Danish Meteorological Institute, Copenhagen, Denmark
[4]Tallinn University of Technology, Tallinn, Estonia
[5]Bundesamt für Seeschifffahrt und Hydrographie, Hamburg, Germany

**Correspondence:** Tuomas Kärnä (tuomas.karna@fmi.fi)

**Abstract.** This paper describes Nemo-Nordic 2.0, an operational marine forecast model for the Baltic Sea. The model is
based on the NEMO (Nucleus for European Modelling of the Ocean) circulation model and the previous Nemo-Nordic 1.0
configuration by Hordoir et al. [Geosci. Model Dev., 12, 363–386, 2019]. The most notable updates include the switch from
NEMO version 3.6 to 4.0, updated model bathymetry and revised bottom friction formulation. The model domain covers the
Baltic and the North Seas with approximately 1 nautical mile resolution. Vertical grid resolution has been increased from 3 to
1 m in the surface layer. In addition, the numerical solver configuration has been revised to reduce artificial mixing to improve
the representation of inflow events. Sea-ice is modeled with the SI3 model instead of LIM3. The model is validated against
sea level, water temperature and salinity observations, as well as Baltic Sea ice chart data for a two-year hindcast simulation.
Sea level root mean square deviation (RMSD) is typically within 10 cm throughout the Baltic basin. Seasonal sea surface
temperature variation is well captured, although the model exhibits a negative bias of approximately -0.5°C. Salinity RMSD is
typically below 1.5 g/kg. The model captures the 2014 Major Baltic Inflow event and its propagation to the Gotland Deep. The
skill analysis demonstrates that Nemo-Nordic 2.0 can reproduce the hydrographic features of the Baltic Sea.
# 1 Introduction
The Baltic Sea is a brackish, semi-enclosed water body in the northern Europe (Figure 1). It has unique characteristics due to
large freshwater input and restricted water exchange with the North Sea. The circulation in the Baltic basin has been simulated
with several numerical models with varying configurations (e.g., Lehmann, 1995; Meier et al., 1999; Funkquist and Kleine,
2007; Berg and Poulsen, 2012; Gräwe et al., 2015a; Hordoir et al., 2019). Due to the complex non-linear interaction with the
North Sea (Gustafsson, 1997; Gustafsson and Andersson, 2001), it is widely accepted that the North and Baltic Seas must be
modeled as a single, coupled system (Daewel and Schrum, 2013; Pätsch et al., 2017; Hordoir et al., 2019).
The North Sea is a marginal sea with open connections to the North Atlantic, and hence it is strongly influenced by tides
and influx of Atlantic water (Huthnance, 1991). Tides are predominantly semidiurnal and propagate cyclonically around the





North Sea; they dissipate in Kattegat. The northern part of the North Sea is stratified throughout the year due to the outflow
of brackish Baltic water (forming the Norwegian Coastal Current) and inflow of Atlantic water through the deep Norwegian
Trench (Winther and Johannessen, 2006).
The shallow Danish Straits region (Fig. 1c) is a transition region between the saline North Sea and the brackish Baltic Sea. In
the Baltic Sea, tides are small ($< 10$ cm) and play a minor role in sea surface height (SSH) dynamics. Short-term SSH variations
are induced by meteorological forcing: atmospherically-induced seiche oscillations are common and can result in large SSH
variations. In addition, SSH is affected by the total water amount in the Baltic Sea, controlled by the riverine freshwater input
and water exchange through the Danish Straits.
In the Baltic Sea, seasonal thermocline starts to develop in late spring and reaches its maximum depth of 10–30 m typically
in late July or early August. The deeper areas of the southern Baltic Sea, Baltic Proper and Gulf of Finland have permanent
stratification, the halocline lying typically at 40–80 m depth. The Gulf of Bothnia, on the other hand, has a relatively weak
salinity stratification: Seasonal mixing reaches depths of 80–90 m and permanent stratification exists only in the deeper parts
of the Bothnian Sea.
During the winter, sea surface temperature (SST) reaches the freezing point in the northern parts of the Baltic Sea and sea
ice forms every year. During average winters, about 45% of the surface area freezes; In severe winters the entire Baltic Sea can
be covered by sea ice (Leppäranta and Myrberg, 2009).
The salt balance in the Baltic Sea is controlled by freshwater discharge, vertical mixing, and episodic salt water inflows
through the Danish Straits. The Major Baltic Inflow (MBI) events bring large quantities of saline water into the Baltic Sea.
Large MBIs occur roughly once per decade (Mohrholz, 2018) and are important for maintaining the salt balance as well as
ventilating the oxygen-depleted deep waters in the Baltic Sea (Mohrholz et al., 2015; Gräwe et al., 2015b). MBIs are driven
by barotropic pressure gradient (sea level difference) between Kattegat and the Arkona basin. The barotropic forcing brings
saline water from Kattegat into the Arkona basin (Gustafsson and Andersson, 2001). From there onward the inflow continues
as a baroclinic bottom current filling the subsequent sub-basins (Gräwe et al., 2015b).
Saline waters can enter the Arkona basin via two routes: Via the Belt Sea, or through the Sound (Fig. 1c). The main inflow
pathway is trough the Great Belt and subsequently the Belt Sea; The final obstruction in this route is the Darss Sill where water
depth is 18 m. The secondary, faster route is via the Sound, separated from the Arkona basin by the Drogden Sill (8 m sill
depth). The volume flux trough the Belt Sea is significantly higher (typically 70% – 80% of total volume flux; Mohrholz et al.
49   2015).

Modeling the circulation in the Baltic Sea is challenging due to the complex topography, strong stratification, and dense
inflows related to the MBIs. Water exchange in the system is governed by the fluxes between the sub-basins. Many constraining
regions, such as the Danish Straits or the Archipelago Sea, are characterized by fine-scale bathymetric features that are difficult
to resolve in operational models. Representing the Danish Straits region, for example, requires sub-kilometer scale horizontal
resolution (She et al., 2007; Gräwe et al., 2015a; Stanev et al., 2018). Similarly, the Archipelago Sea consists of narrow channels
and thousands of small islands that cannot be captured with typical grid resolutions. In practice, resolving the dynamics in these
regions requires nested grids (Gräwe et al., 2015a) or unstructured meshes (Stanev et al., 2018).



Salt pulses propagate as a density-driven bottom current from basin to basin. In circulation models, artificial numerical
mixing can slow down or completely arrest the propagation of the current; Numerical mixing dilutes the water masses, reducing
the horizontal pressure gradient as well as stratification. Spurious vertical mixing can also erode the permanent halocline in
deeper basins, leading into excessive ventilation of the oxygen-depleted deep waters. In finite volume models, the accuracy
of the advection scheme has a significant impact on the level of numerical mixing Zalesak (1979); Hourdin and Armengaud
(1999); Lévy et al. (2001). Previous studies suggest that vertical grid resolution plays an important role in retaining the salt
pulse density structure in the Baltic Sea (Hofmeister et al., 2011; Gräwe et al., 2015a).
Operational ocean modeling has a fairly long history in the Baltic Sea, starting already in the mid 1990s with HIROMB (High
Resolution Operational Model for the Baltic). It was a cooperation involving many Baltic institutes who gathered around a
common circulation model with the same name. The cooperation itself later became the modeling part of the Baltic Operational
Oceanographic System (BOOS; https://boos.org; She et al. 2020), or the BOOS Modelling Programme. The HIROMB model
itself existed in several branches in different institutes for many years, e.g. the HIROMB model (Funkquist and Kleine, 2007;
Axell, 2013) at the Swedish Meteorological and Hydrological Institute (SMHI), and the HIROMB BOOS Model (HBM; Berg
and Poulsen 2012). The first version of a common circulation model built around the Nucleus for European Modelling of the
Ocean (NEMO) was called Nemo-Nordic 1.0 and was described and validated by Pemberton et al. (2017) and Hordoir et al.
(2019). It was based on NEMO version 3.6 and was coupled to the integrated ice model LIM3 (Vancoppenolle et al., 2009).
In this paper, we present an updated Nemo-Nordic 2.0 operational forecast system for the Baltic Sea based on the NEMO
circulation model version 4.0 (Madec et al. 2019). The model domain covers the North and Baltic Seas (Fig. 1). The setup is
based on the Nemo-Nordic 1.0 configuration (Hordoir et al., 2019). Compared to Nemo-Nordic 1.0, most notable updates are
the switch from NEMO 3.6 to NEMO 4.0, updated bathymetry and revised bottom friction formulation. Vertical grid resolution
has been increased in the surface layer from 3 to 1 m. We have also revised the numerical schemes (e.g. advection of momentum
and tracers) to reduce artificial mixing. Finally, NEMO 4.0 uses the SI3 sea ice model instead of LIM3.
The aim of this article is to validate the operational Nemo-Nordic 2.0 forecast system. The model configuration is identical
to the operational EU Copernicus Marine Service forecast model for the Baltic Sea. The validation is based on a 2-year
hindcast simulation that uses similar forcing as in the operational configuration. The presented validation run uses a 3 km DMI
HIRLAM atmospheric forcing whereas the operational model will use 2.5 km MetCoOP HARMONIE model forecast padded
with ECMWF HRES forecast data in North Sea regions outside the MetCoOP domain.
The Nemo-Nordic 2.0 model skill is assessed with respect to SSH, water temperature and salinity, as well as sea ice coverage
using observations from tide gauges, FerryBox instruments, vertical profiles, as well as ice charts. The model configuration
and used observation data sets are presented in Section 2. Section 3 presents the model skill metrics, followed by a discussion
and conclusions in Section 4.

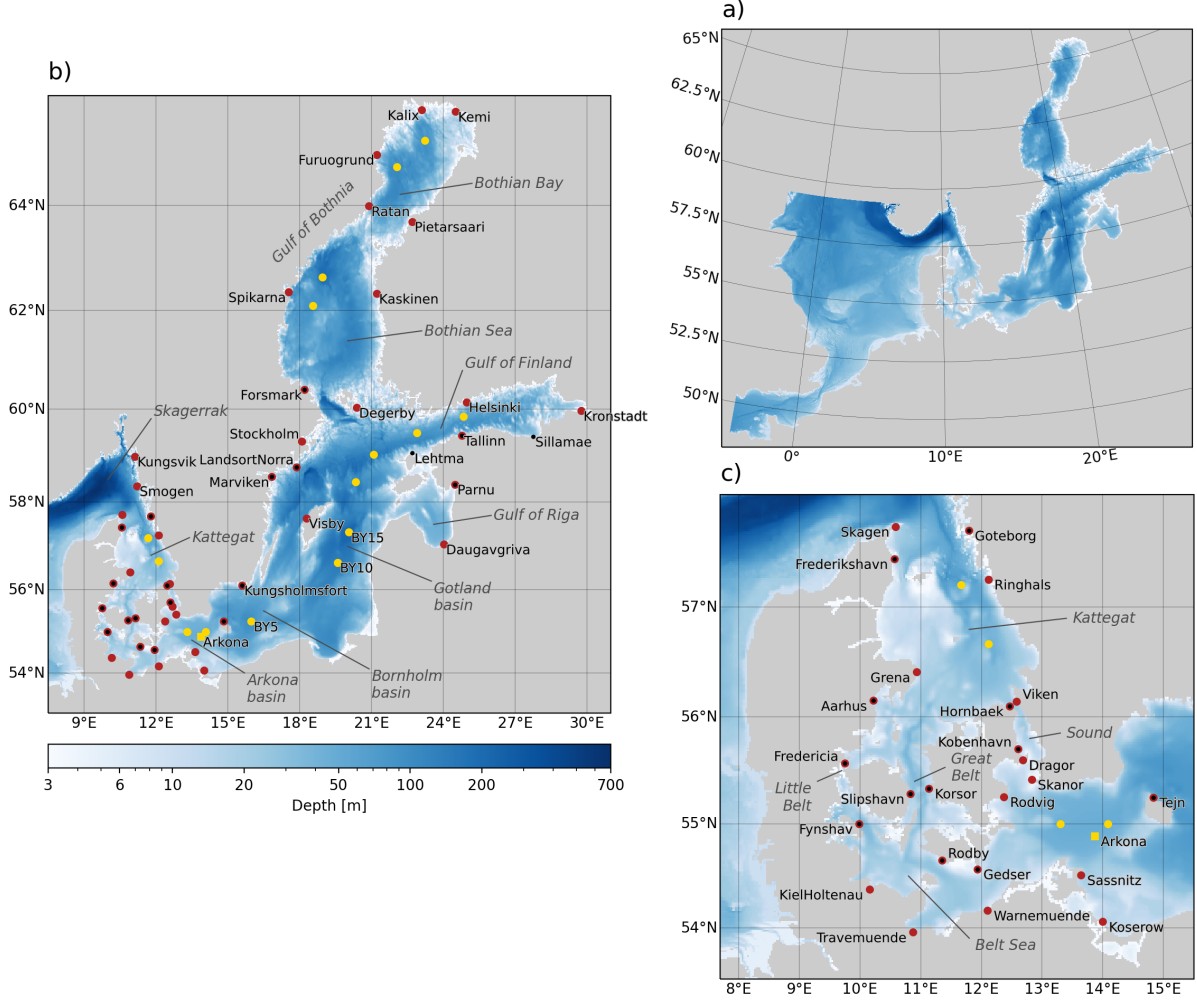

**Figure 1.** Computational domain and bathymetry; a) the entire model domain, b) the Baltic Sea, c) Danish Straits region. Red and black dots indicate station locations with SSH and SST observations, respectively; Yellow dots indicate vertical profile locations.

## 2  Methods

### 2.1  Model domain and configuration

The model setup is based on the Nemo-Nordic 1.0 configuration (Hordoir et al., 2019) and NEMO version 4.0 (Madec et al. 2019; subversion repository revision 11281). The grid covers the North and Baltic Seas, spanning from 4.15278°W to 30.18021°E and 48.4917°N to 65.8914216°N (Figure 1 a). The grid spacing is 0.0277775° and 0.0166664° in the zonal and meridional directions, respectively, resulting in approximately 1 nautical mile resolution. In the North Sea, the open boundaries are located in the western part of the English Channel and between Scotland and Norway.



The model uses a $z^*$ grid in the vertical direction, consisting of 56 levels. The layer thickness is 1 m at the surface, increasing
to 10 m at 75 m depth, and maximum 24 m at 700 m depth. In the bottom cell a partial step formulation is used, i.e. the location
of the bottom node is fitted to the local bathymetry instead of fixing it to the full $z$ level height.
The model's bathymetry is derived from the global GEBCO data set (the GEBCO-2014 grid, version 20150318; http://www.
gebco.net, last access: March 29, 2021). The data was interpolated to the centroids of the model grid. The land mask was
generated from OpenStreetMap coastline data (https://www.openstreetmap.org, last access: March 29, 2021) and the GEBCO
bathymetry. In the Baltic Sea, the minimum depth was set to 3 m. In the North Sea, the minimum depth varies between 5 and
10 m to accommodate tidal variations as wetting and drying is not taken into account. The bathymetry was modified along
the west coast of Denmark by masking out shallow regions (such as the Limfjord area) to improve the propagation of tides to
Skagerrak and Kattegat. Furthermore, to improve the inflow of salt water into the Baltic Sea, the bathymetry was modified in
the Danish Straits region (see Sect. 2.2).
The model configuration was tuned to accurately simulate surface gravity waves and internal gravitational currents. We use
a non-linear free surface formulation with mode-splitting. The baroclinic and barotropic time steps are 90 and 3 s, respectively.
Fast variations in the 2D fields are filtered out by averaging over two baroclinic time steps. A vector-invariant form of the mo-
mentum equation is used with an energy and enstrophy conserving advection scheme. This choice improves the representation
of baroclinic eddies, reduces noise in the velocity field and also improves the interaction of currents and topography in partial
step configurations (Madec et al., 2019).
Bottom friction is imposed with implicit nonlinear log-layer parameterization. The bottom drag, $C_d$, was computed from a
spatially variable bottom roughness length parameter, $z_0^b$,
$$C_d = \left( \frac{\kappa}{\log\left(\frac{z_0^b + h_0/2}{z_0^b}\right)} \right)^2,$$
(1)

where $h_0$ is height of the bottom cell, and $\kappa$ is the von Karman constant. As NEMO 4.0 only allows users to specify the $C_d$ field
directly, the formulation (1) was implemented in NEMO, introducing a user-defined $z_0^b$ field. With the $z^*$ vertical coordinates,
the cell height varies in time, and consequently $C_d$ becomes time dependent. The bottom roughness length formulation (1)
is consistent with the law of the wall boundary layer theory, and it is the preferred parameterization especially in coastal and
shallow regions. The additional benefit is that, unlike $C_d$, $z_0^b$ does not depend on the cell height, making the configuration more
robust with respect to changes in the vertical grid and bathymetry.
The bottom roughness field was tuned to improve sea level skill throughout the model domain. In the English Channel $z_0^b$
was set to 0.3 mm to avoid dampening the tidal signal along the continental coast. In the northwestern part of the North Sea
$z_0^b = 1$ cm and $z_0^b = 3$ cm in the Danish Straits to introduce sufficient dissipation. In the Baltic Sea, 1 mm was used to prevent
excessive damping of seiche motions.
Vertical turbulence is modeled with the Generic Length Scale model (Umlauf and Burchard, 2003; Reffray et al., 2015),
using $k - \varepsilon$ closure and Canuto A stability functions. Horizontal diffusion of momentum and tracers were modeled with a
Laplacian formulation with constant-in-time diffusivity and viscosity. In the surface layer (top 10 m), viscosity and diffusivity





were set to $50$ and $5$ m$^2$ s$^{-1}$, respectively. For the rest of the water column, horizontal viscosity was $0.01$ m$^2$ s$^{-1}$ and diffusivity
was neglected to avoid excessive mixing in the bottom layer (Hordoir et al., 2019).
Sea-ice dynamics are modeled with the SI3 sea-ice model. SI3 solves the sea-ice thermodynamics, advection, rheology and
ridging/rafting. The landfast ice parametrization by Lemieux et al. (2016) is used. The model consist of five ice categories and
one snow category. Ice thickness categories are defined with the thickness bounds 0.45, 1.13, 2.14, and 3.67 m. In this work,
we use the standard settings in the sea-ice model, originally developed for the global ocean, for the ice thickness categories;
Further tuning of the configuration for the Baltic Sea will be performed at a later stage.
In NEMO 4.0, we use the TEOS-10 equation of state (IOC, 2010; Roquet et al., 2015) to compute water density. Conse-
quently, the modeled water salinity and temperature are in Absolute Salinity (g/kg) and Conservative Temperature (°C) units,
respectively. When comparing against observations, the model's temperature is converted to in-situ temperature.

## 2.2   Improving salinity inflows

Salt water inflows to the Baltic Sea are extremely sensitive to bathymetric features, especially in the Danish Straits region, and
numerical aspects of the model. To improve the representation of the inflow events, the bathymetry in the Danish Straits area
was tuned. It is known that the narrow channels in the Straits (the Little Belt, Great Belt, and the Sound; Fig. 1c) play a crucial
role in the water exchange between the North Sea and the Baltic Sea by allowing dense waters to creep to the Arkona basin.
These channels are, however, very narrow and resolving them properly would require a much finer grid resolution (approxi-
mately 250 m; e.g., Stanev et al. 2018) than what is feasible in an operational model of the presented extent. Consequently, in
coarser resolution configurations, the bathymetry must be tuned to facilitate the influx of dense water masses (e.g., She et al.,

146   2007).

First, the bathymetry in the Great Belt was artificially deepened by a factor of 1.3 to allow the influx of dense waters. Next,
the bathymetry was smoothed in the Danish Straits region between Kattegat and the Arkona basin by applying a Gaussian
filter on the bathymetry raster field (using standard deviation $\sigma$ of 2 grid cells). Large local gradients in the bathymetry field
introduces local obstructions ("sills") and also generates noise in the velocity field which tends to mix the tracers and thus
reduce the pressure gradient driving the gravitational current. A less intrusive smoothing (a Gaussian filter with $\sigma = 0.66$) was
applied in the rest of the Baltic basin; our test runs indicate that a smoother bathymetry improves otherwise underestimated sea
level variability in coastal regions, e.g. in the Gulf of Bothnia. Both the smoothing and deepening of the local bathymetry were
necessary to reproduce Major Baltic Inflow events in the model.
In finite volume models, the accuracy of tracer and momentum advection schemes has a great impact on the effective
numerical dissipation of the model. In the baroclinic regime, numerical models tend to generate noise at grid scale in the
velocity field which tends to increase artificial mixing of tracers. Griffies and Hallberg (2000) and Ilıcak et al. (2012) have
demonstrated that adding a suitable amount of viscosity can effectively suppress such oscillations in the velocity field, and
therefore reduce the overall mixing of tracers. Moreover, higher order advection schemes can be utilized to reduce numerical
mixing but they can also generate spurious, unphysical, oscillations in the advected quantity. In this work, we have chosen to
use the 3rd order Upstream Biased Scheme (UBS), and the 4th order centered scheme, for horizontal and vertical advection





of momentum, respectively. The same advection scheme combination is used in ROMS (Regional Oceanic Modeling System;
Shchepetkin and McWilliams 2005) as well. The UBS scheme adds some dissipation in the high-frequency part which reduces
noise in the velocity field. The 4th order vertical scheme, on the other hand, is less dissipative and can retain sharp gradients
better (e.g. in a stratified two-layer flow). For tracers, we use the 4th order Flux Corrected Transport (FCT) scheme in both
horizontal and vertical directions; the vertical scheme uses the COMPACT formulation. The 4th order FCT schemes ensure
lower numerical dissipation while being positive definitive, i.e. they do not generate any spurious overshoots in the tracer fields.
Switching to these higher-order advection schemes significantly improved the magnitude of salt inflow to the Arkona basin
(not shown).
Our tests also indicated that the vertical eddy diffusivity from the turbulence model caused excessive vertical mixing in
the Belt Sea–Arkona region, effectively stopping the propagation of the inflows. As a remedy, we lowered the Galperin limit
parameter in the $k - \varepsilon$ model to a value 0.10. In contrast, the value 0.17 was used in Nemo-Nordic 1.0 (Hordoir et al., 2019).

## 173 2.3 Forcings

The simulation covers a 2-year period from October 1, 2014 to September 30, 2016. Initial conditions for water temperature
and salinity were obtained from a 14-month spin-up run (August 1, 2013 – September 30, 2014); SSH and velocity were
initialized as zero. Because the model is initialized at rest, the first month of the simulation is excluded from the analysis and
the model skill is evaluated for the remaining 23 months (November 1, 2014 to September 30, 2016).
The model is forced with the HIRLAM atmospheric forecast model data (http://hirlam.org, last access: March 29, 2021).
The 10 m wind, 2 m air temperature, 2 m specific humidity, incoming long and short wave radiation, total precipitation, solid
precipitation, and surface atmospheric pressure fields at 1 h temporal resolution are fed to the NEMO model. We use the NCAR
bulk formulae (Large and Yeager, 2004) of the Aerobulk package (Brodeau et al., 2016) to evaluate the turbulent air–sea fluxes.
In addition, atmospheric pressure gradient is applied in the momentum equation. To account for slightly underestimated wind
speeds in the atmospheric model, the wind stress was multiplied by a factor 1.1 in NEMO.
Along the open boundaries in the North Sea, SSH, depth-averaged velocity, as well as vertical profiles of temperature and
salinity are prescribed from the CMEMS Northwestern Shelf forecast model (Graham et al., 2018). This configuration is
sufficient to prescribe the tidal signal in the North Sea, and sub-tidal variation of SSH, temperature and salinity.
River forcing data are obtained from the EHYPE model (Arheimer et al., 2012). River discharge and water temperature are
prescribed at 729 rivers along the coasts with 1 day temporal resolution. Salinity of riverine water is set to zero.

## 189 2.4 Observations

Observational data was obtained from the Copernicus Marine Environment Monitoring Service (CMEMS) near-realtime, in-
situ observation catalog (`INSITU_BAL_NRT_OBSERVATIONS_013_032`). SSH data was obtained from 45 tide gauges
across the whole Baltic basin (red markers in Fig. 1). Tide gauge SST observations were scarcer, focused more on the southern
part of the basin (black markers). Vertical profile data of water temperature and salinity were obtained at locations shown
with yellow markers. Only stations with more than 6 profiles in the study period were included in the analysis. In addition,





continuous vertical profile observations from the Arkona buoy (indicated by a yellow square in Fig. 1) were used; The data
contain temperature and salinity observations at 1 h temporal resolution from eight CTDs at different depths (2 to 43 m).
FerryBox surface temperature and salinity observations were included from two ferries, TransPaper, and FinnMaid. As the
observations have high sampling rate (typically $< 30$ s), the data was binned to 10 min temporal resolution. The binned data
consists of mean temperature, salinity, and ship location during the 10 min time window. Furthermore, the data was manually
quality checked to remove spurious values (such as abnormally long periods showing constant salinity or temperature).
The salinity observations are in practical salinity units; for model skill analysis, observed salinity was converted to Absolute
Salinity units. Prior to computing the error metrics, the model data was interpolated to the observation locations and time
stamps. Unless otherwise specified, spatial interpolation was carried out with nearest neighbor search while linear interpolation
was used in time.
Sea ice extent was computed from digitized ice charts by the Finnish Meteorological Institute. The ice chart frequency varied
between 1 and 7 days in the study period. In the beginning of the ice season when ice coverage is scarce, ice charts are usually
generated at 3 or 4 day intervals. Daily charts are typically available from late December onward.

## 208 2.5 Error metrics

The model skill is quantified with standard statistical measures. Let $o_i$ and $m_i$, $i = 1, \ldots, N$ be the observed and modeled time
series, respectively. Denoting the mean of the time series by $\overline{m}$, the bias, Root Mean Square Deviation (RMSD) and Centered
Root Mean Square Deviation (CRMSD) are defined as
$\mathrm{BIAS} = \overline{m} - \bar{o},$
$\mathrm{RMSD}^2 = \dfrac{1}{N} \sum\limits_{i=1}^{N} (m_i - o_i)^2,$
$\mathrm{CRMSD}^2 = \dfrac{1}{N} \sum\limits_{i=1}^{N} ((m_i - \overline{m}) - (o_i - \bar{o}))^2.$
Standard deviation ($\sigma_m$) and correlation coefficient ($R$) are given by
$\sigma_m^2 = \dfrac{1}{N} \sum\limits_{i=1}^{N} (m_i - \overline{m})^2,$
$R = \dfrac{1}{\sigma_o \sigma_m} \dfrac{1}{N} \sum\limits_{i=1}^{N} (m_i - \overline{m})(o_i - \bar{o}).$
CRMSD is related to $\sigma_m$ and $R$ though the equation,
$\mathrm{CRMSD}^2 = \sigma_o^2 + \sigma_m^2 - \sigma_o \sigma_m R,$     (2)





which can be visualized in a Taylor diagram (Taylor, 2001). In this work, we are normalizing the Taylor diagram by scaling
the variables with $\sigma_o$:
$$\text{NCRMSD}^2 = 1 + \sigma_m'^2 - \sigma_m R, \tag{3}$$
$$\text{NCRMSD} = \frac{1}{\sigma_o}\text{CRMSD},$$
$$\sigma_m' = \frac{\sigma_m}{\sigma_o},$$
where NCRMSD and $\sigma_m'$ are the normalized CRMSD and standard deviation of the model, respectively. Normalization leads
into dimensionless metrics and permit comparison of different data sets in a single figure. As the Taylor diagram does
not contain the bias, SST comparisons also include a target diagram depicting normalised bias, $\text{NBIAS} = \text{BIAS}/\sigma_o$, and
NCRMSD. In the target diagram, NCRMSD has been augmented with the sign of $\sigma_m - \sigma_o$. We also use normalized RMSD,
$\text{NRMSD} = \text{RMSD}/\sigma_o$. NRMSD is a useful dimensionless metric: The value 1.0 can generally be regarded as a threshold for
poor model skill. Indeed, setting $m_i$ to the mean value of the observations results in NRMSD=1.0.
Comparing simulated SSH to observations is challenging. The exact vertical reference datum of the circulation model is not
well defined. In addition, the available tide gauge data originates from different institutes that use different vertical datums.
Although the observations can be converted to a common reference level, the modeled SSH is not directly comparable in terms
of the mean elevation.
Simulated SSH depends on the forcing at the open boundaries, datum of model's bathymetry, riverine freshwater flux, and
water density. At the open boundaries, we are using SSH from the Northwestern Shelf AMM model Graham et al. (2018). Its
vertical reference is close to the mean sea level, but not exactly defined. Likewise, the GEBCO bathymetric data is nominally
defined against local mean sea level, but in practice it consists of several local bathymetric datasets whose vertical datum may
differ. Thus the modeled mean water depth and SSH may exhibit local biases, especially in coastal regions.
Because SSH bias is not a reliable model skill metric, SSH performance is therefore assessed with centralized metrics, e.g.
CRMSD and Taylor diagrams.

## 242 3 Results

### 243 3.1 Sea surface height

SSH error metrics (CRMSD and standard deviation) are shown in Fig. 2. The model skill is generally good, CRMSD being
below 10 cm at most stations. For the Baltic basin (Gulf of Bothnia, Gulf of Finland, Archipelago Sea, Gotland, Bornholm and
Arkona basins), CRMSD is below 8 cm indicating that seiche waves are well reproduced. The only exception is Kronstadt,
located at the eastern end of Gulf of Finland, where CRMSD is 10 cm. The deviation is generally larger in the Danish Straits
and Kattegat/Skagerrak region where tidal variations are significant. The largest errors occur at Fredericia, in the Little Belt
region, where CRMSD exceeds 12 cm. The model skill in these locations is affected by the Little Belt topography which is
difficult to resolve at 1.8 km resolution; the strait itself is less than 1 km wide in the narrowest part. The model resolution

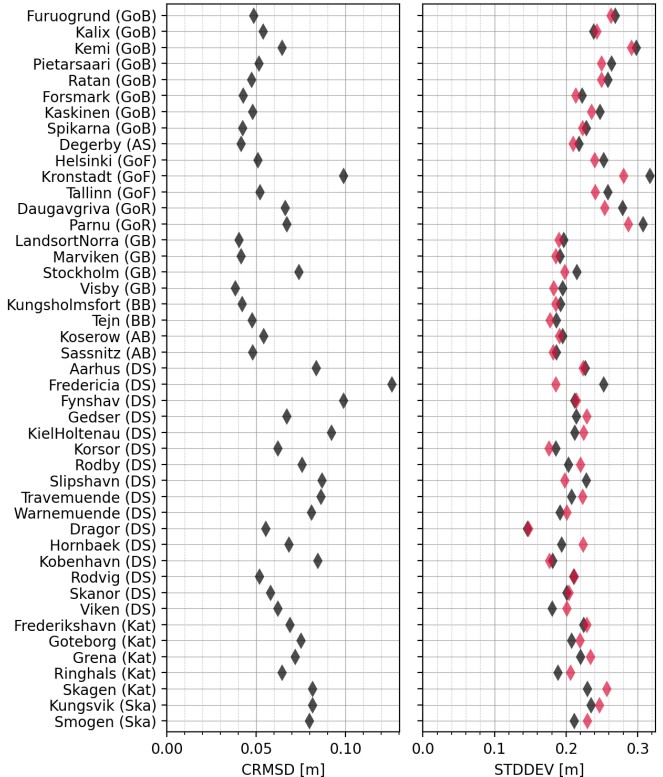

**Figure 2.** Sea surface height error metrics. Black and red symbols denote the model and observations, respectively. Sub-basins are indicated by the abbreviations: GoB, Gulf of Bothnia; AS, Archipelago Sea; GoF, Gulf of Finland; GB, Gotland Basin; BB, Bornholm Basin; AB, Arkona Basin; DS, Danish Straits; Kat, Kattegat; Ska, Skagerrak.

also affects the skill in other locations such as Kobenhavn in the Sound, and stations along the German coast of the Belt Sea (KielHoltenau, Warnemuende, Travemuende). All of these tide gauges are located inside an estuary mouth or harbor area which the present model cannot resolve.

In general, the model reproduces SSH standard deviation well (Fig. 2). The difference is within 2 cm in most cases; the largest deviation at Fredericia is approximately 7 cm. In the Baltic basin, the model has a tendency to overestimate variability. In the Danish Straits, the variability is slightly underestimated at several locations although overestimation also occurs.

The Taylor diagram (Fig. 3) shows that the model reproduces SSH variability well. Most stations have NCRMSD < 0.45, and the correlation coefficient is generally above 0.90. The deviations is the smallest in the Baltic basin where NCRMSD < 0.30 (except at Kronstadt and Stockholm) and the correlation coefficient is above 0.95 (except at Stockholm). The model performance at Stockholm is likely affected by the archipelago which is not fully resolved in the model. The Taylor diagram confirms that deviation is larger in the Danish Straits region; the worst stations are Fredericia in the Little Belt, as well as Kobenhavn.

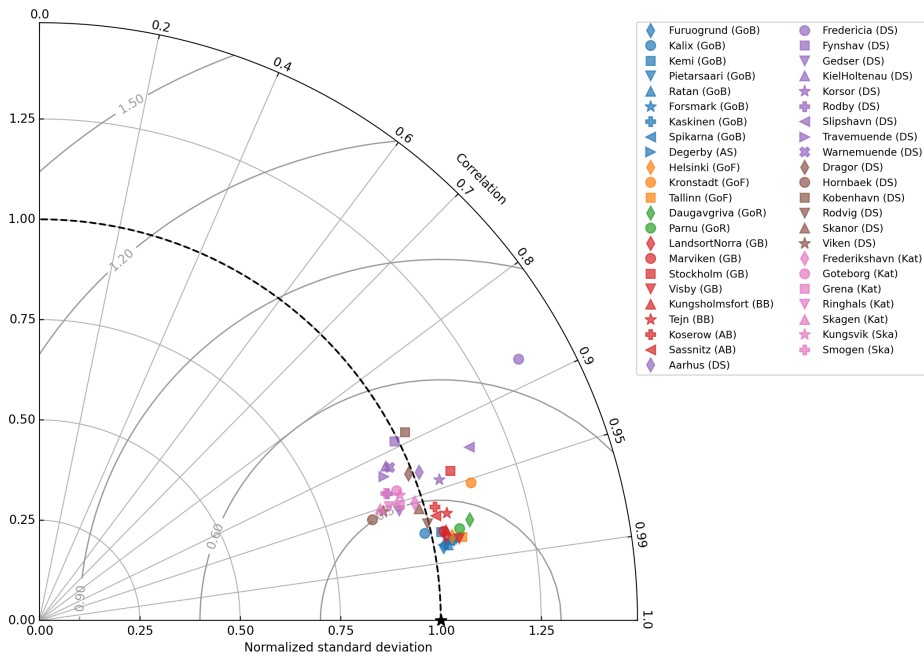

**Figure 3.** Taylor diagram of sea surface height error metrics. The metrics have been normalized by the standard deviation of the observations. The sub-basin abbreviations are as in Fig. 2.

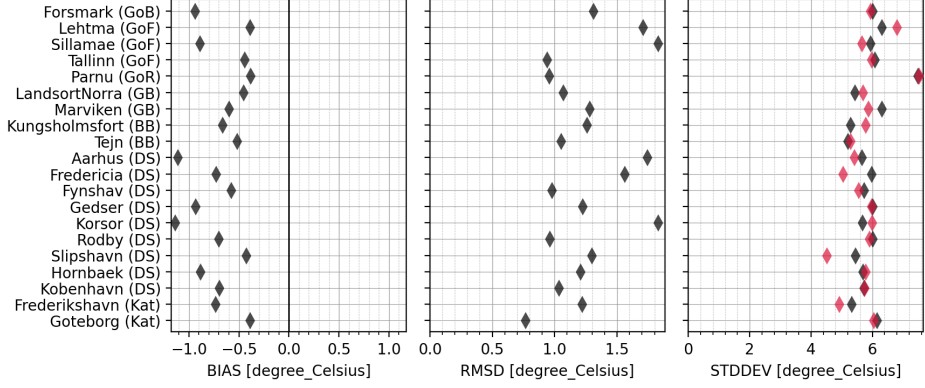

**Figure 4.** Surface temperature error metrics. Black and red symbols denote the model and observations, respectively. The sub-basin abbreviations are as in Fig. 2.

## 3.2 Sea Surface Temperature and Salinity

Sea surface temperature (SST) is mostly governed by surface heat fluxes, driven by the seasonal cycle of solar radiation and air temperature, and vertical mixing. In addition, near the coast (where tide gauges are located), river discharge temperature



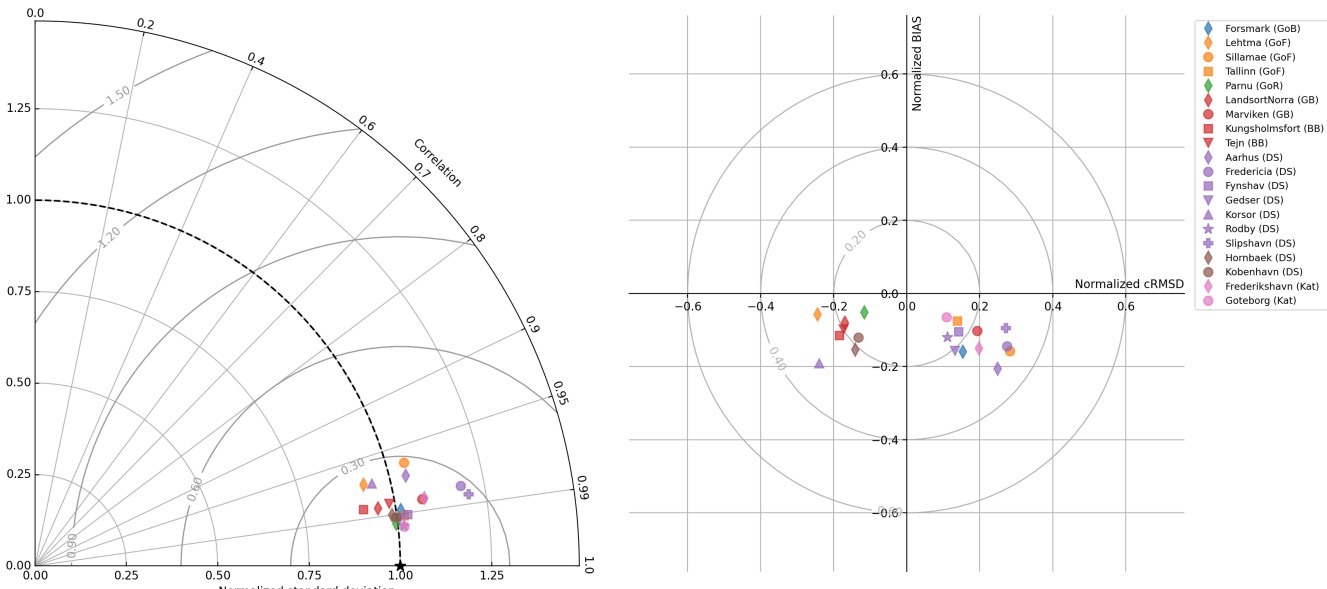

**Figure 5.** Taylor and target diagram of surface temperature error metrics. The metrics have been normalized by the standard deviation of the observations. The sub-basin abbreviations are as in Fig. 2.

can cause local variations. During summer, SST in shallow coastal regions can exceed 20°C. In the northern Baltic Sea, SST
regularly reaches freezing point during winters.
Tide gauge SST metrics are shown in Fig. 4. In general, there is no clear pattern across the domain. The model has a
negative bias (typically between –0.4 and –0.9°C) at almost all stations; the largest bias (exceeding –1.1°C) occurs at Korsor.
The RMSD is below 1.9°C in all cases. The standard deviation is typically close to the observed value.
The Taylor diagram (Fig. 5, left) shows that SST skill is good in general. All locations are within 0.30 NCRMSD, and have
a correlation coefficient above 0.95. The normalized standard deviation is generally close to unity except at Fredericia and
Slipshavn. The model skill indicates that the model captures the seasonal SST variability well. The normalized target diagram
(Fig. 5, right) shows that the negative bias is quite small (typically NBIAS < 0.20) compared to the overall SST variability (i.e.
seasonal cycle).
Comparison against FerryBox SST from the TransPaper and FinnMaid ferries are shown in Figures 6 and 7, respectively (top
row). Although the FerryBox data has sparser temporal coverage than the tide gauge observations, it is useful for validating
the modeled SST in open waters away from the coasts. Also here the annual temperature cycle is well reproduced. The model
bias is less than –0.5°C for the two ferries (Fig. 6c and 7c). RMSD is below 0.9°C, suggesting that SST performance is better
in the open sea than at the tide gauge locations.
The TransPaper comparison shows a negative bias in the Baltic Proper and Gulf of Bothnia in summer 2015 (June–July; Fig.
6 c). The bias is smaller during the following fall (October–December). A negative bias is visible in summer 2016 as well. The

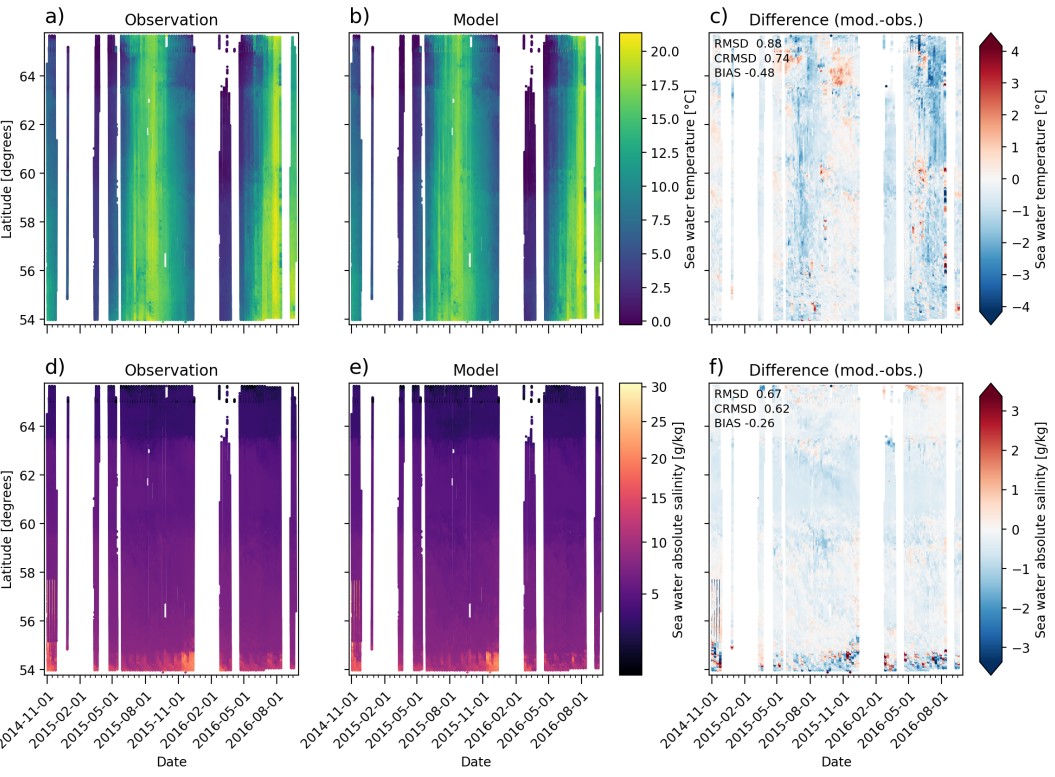

**Figure 6.** Surface temperature and salinity comparison against TransPaper FerryBox observations. The ferry operates between Oulu (top) and Lübeck (bottom). In the beginning of the data set (November 2014) the ferry also visited Gothenburg.

shorter FinnMaid data set also shows the negative bias during summer 2015 (Fig. 7 c). Although the data coverage is sparse,
the comparison therefore suggest that the model has a negative bias during summer months (June–July) whereas the bias is
smaller in fall. The magnitude of these deviations is, however, typically below 1.5°C.
FerryBox Sea surface salinity (SSS) comparison is shown in Figures 6 and 7 (bottom row). TransPaper observations (Fig.
6d) clearly show the SSS gradient ranging from roughly 15 g/kg in the Belt Sea to 0 g/kg in the northern part of the Bothnian
Bay. In November 2014, the ferry also visited Gothenburg where SSS can reach 30 g/kg. The model reproduces the salinity
gradient well, bias is -0.26 g/kg and RMSD is 0.67 g/kg; In the FinnMaid data set (Fig. 7f), the bias and RMSD are similar,
-0.16 and 0.55 g/kg, respectively. In both cases, the deviation is the largest in the Belt Sea and Kattegat where the model
predominantly underestimates SSS. The larger error magnitude is due to the significant salinity gradients and their temporal
variability in this region. The model has a small negative bias (-1 g/kg) in the Bothnian Sea (Fig. 6f). A negative bias is seen in
the Gulf of Finland as well (around latitude 60°N; Fig. 7f).



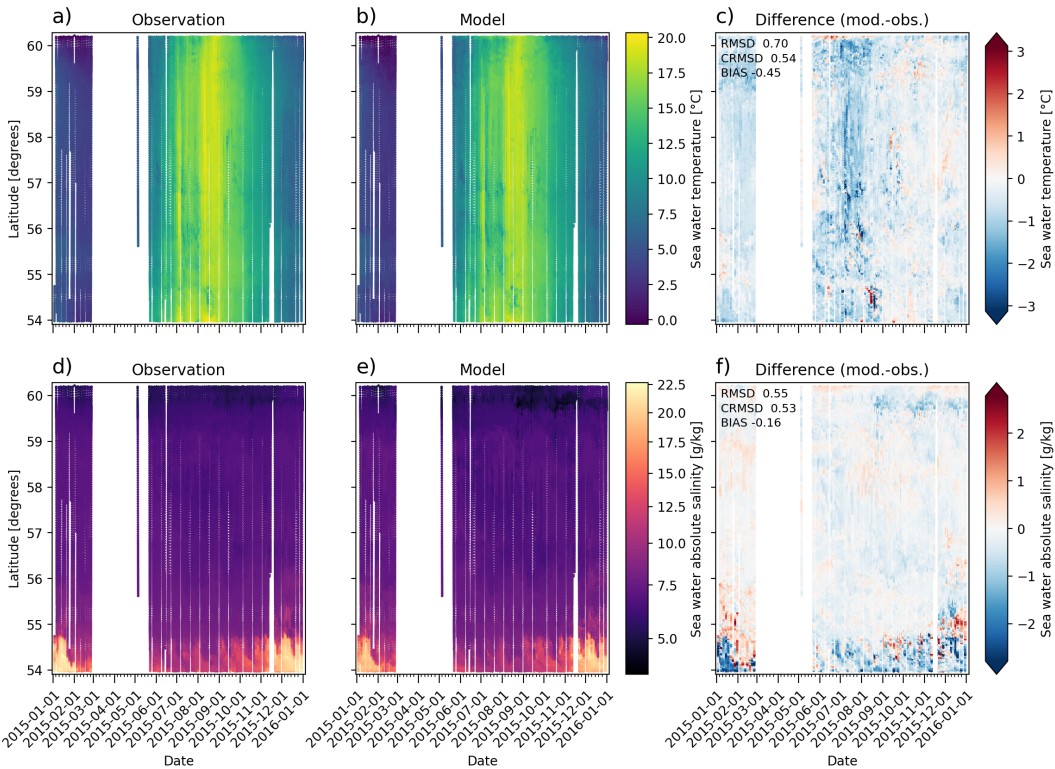

**Figure 7.** Surface temperature and salinity comparison against FinnMaid FerryBox observations. The ferry operates between Helsinki (top) and Lübeck (bottom).

## 3.3 Vertical profiles

Figure 8 shows comparison of temperature and salinity profiles against research vessel observations. In all panels, the outer ring denotes NRMSD, the middle ring RMSD, and the center dot the BIAS. RMSD and BIAS are visualized using the same color map; in all cases light color indicates small deviation. In addition, the metrics across all the plotted stations are printed in the upper left corner.

The temperature profile skill (Fig. 8a) is quite similar throughout the Baltic Sea. NRMSD is below 0.5 at all locations (except at C3 where NRMSD is 0.54) indicating good skill. The metrics for surface temperature (top 10 m; Fig. 8b) show low deviations. The NRMSD of the combined data set is 0.14 and bias is -0.51°C. The high surface temperature skill is consistent with SST results presented in Sect. 3.2: the model can capture the seasonal surface temperature variability driven by the solar radiation. The profile comparison confirms that the skill is good in the entire surface layer.

The bottom temperature skill is poorer (Fig. 8c). RMSD in the Gotland basin is quite low (below 0.5°C) while being higher in the Gulf of Finland and Gulf of Bothnia (0.6 to 1.3°C) In these regions, however, NRMSD exceeds value 2.0 at several stations: While RMSD is moderate, the deviation is significant compared to the low variability of bottom temperature. The bias



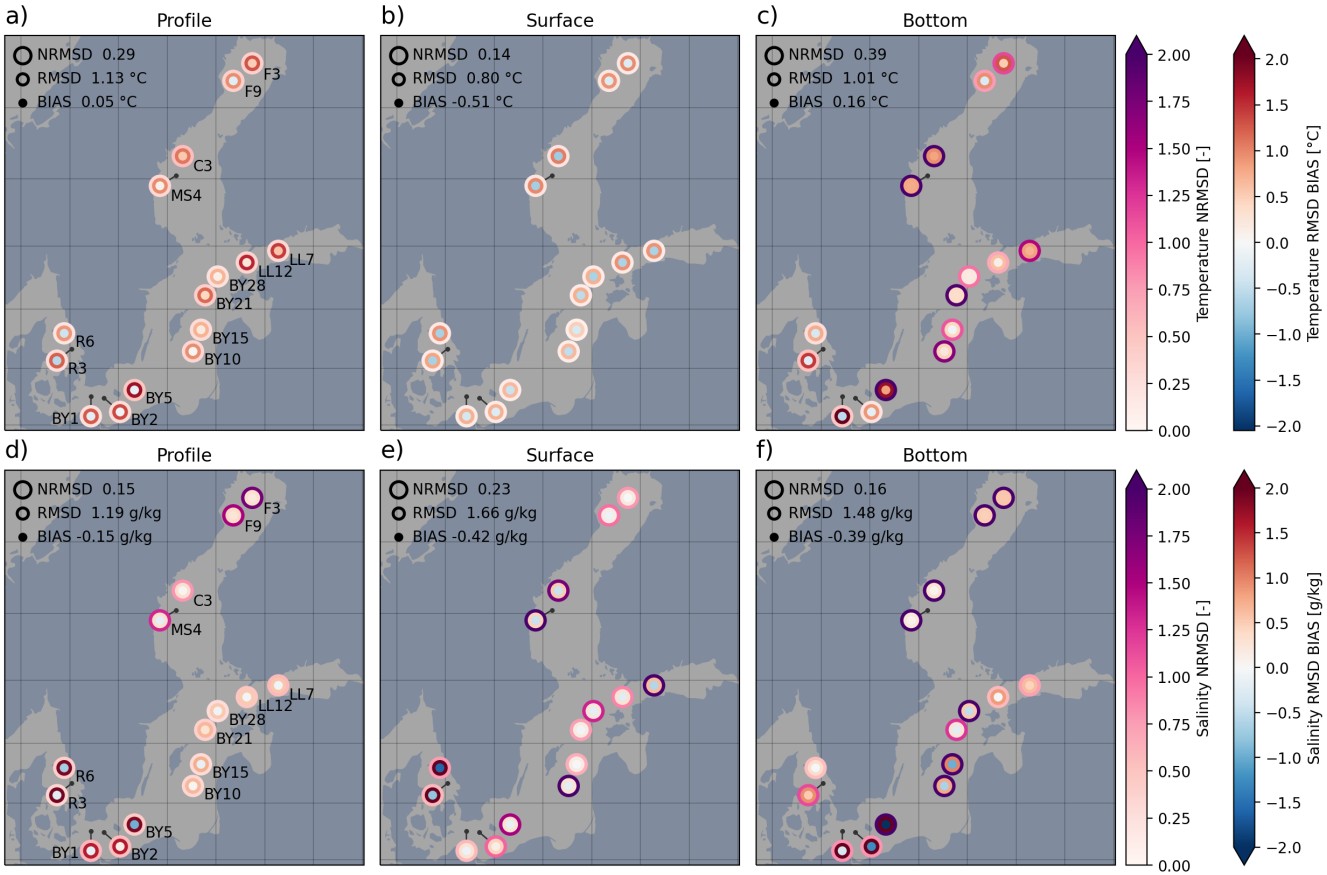

**Figure 8.** Vertical profile comparison against vessel observations; Temperature (top row) and salinity (bottom row). The dots depict NRMSD (outer ring), RMSD (inner ring), and BIAS (center dot). The metrics have been calculated for the entire profile (a,d), surface 10 m (b, e) and bottom (c, f). Combined metrics over all the stations is printed in each panel. Bottom values were computed for the lowest 15% of the water column.

tends to be positive in the Baltic basin, and negative in Kattegat and the Arkona basin. The largest deviation occurs at BY1
where RMSD reaches 2.0°C.
The model reproduces salinity profiles relatively well (Fig. 8d). The deviation is small (RMSD < 0.73 g/kg) in the Gotland
basin and Gulf of Finland. NRMSD is large in the Gulf of Bothnia due to the fact that salinity is low in this region as a result of
riverine freshwater input. In contrast, in the Danish Straits and Kattegat NRMSD is small while RMSD is relatively high (up
to 1.9 g/kg) due to the opposite reason: In this region, salinity regularly varies by more than 10 g/kg.
Surface salinity metrics (Fig. 8e) are generally similar to the whole profile metrics, except NRMSD is high in the Bothnian
Sea, Gulf of Finland, and southern Gotland basin.





The bottom salinity skill is presented in Fig. 8f. NRMSD regularly exceeds 2.0 in the Baltic basin because the variability of
bottom salinity is small. A significant negative bias is visible in the Arkona and especially Bornholm basin; the bias is negative
also in the Gotland basin. The deviation is moderate in Kattegat and the Gulf of Finland. While absolute deviation is small
in the Gulf of Bothnia, NRMSD is large. In general, the results indicate that bottom salinity is underestimated in the Arkona,
Bornholm, and Gotland basins. The skill is poorest in Bornholm (BY5). At this location, the model reproduces the halocline
correctly but tends to underestimate bottom salinity. During the first simulated year the deviation is roughly –2 g/kg; after
November 2015 it is –4 g/kg.

### 3.4 The 2014 Major Baltic Inflow event

A Major Baltic Inflow (MBI) event occurred in December, 2014. Mohrholz et al. (2015) divided the event into four phases:
During the outflow period (November 7 – December 3 2014) strong easterly winds pushed surface waters from the Baltic
Sea to Kattegat, lowering the mean sea level at Landsort by 57 cm. After the winds ceased for a couple of days, the inflow
period was characterized by strong westerly winds. The precursory period (December 3 – December 13, 2014) brought saline
waters through the Sound and into the Belt Sea. On December 13, the saline inflow had reached Darss Sill. At the buoy, the
salinity exceeded 17 g/kg in the entire water column, marking the beginning of the main inflow period. The main inflow period
lasted until December 25, 2014, when the westerly winds, and hence the barotropic inflow, ceased. During this time, the mean
sea level at Landsort rose 95 cm from the lowest value, until the dense saline water mass had reached the Arkona basin. In
the post-inflow period (starting on December 25, 2014), the saline water mass crept further into the Bornholm basin, and the
following downstream sub-basins, driven by the baroclinic pressure gradient (density difference).
The observed salinity from the Arkona Buoy is shown in Figure 9 (panel a). The main salt pulse arrives at the buoy on
December 16, 2014. On December 20, 2014, the dense water reaches 16 m depth. The model replicates the arrival of the
pulse on December 16, 2014 and the main inflow phase (December 20 to December 26, 2014; Figure 9b). The bottom salinity
is underestimated during the onset of the pulse (December 12 to 20 2014) from January 4 2015 onward (Figure 9c). Most
notably, the observations show a secondary salt pulse in April 2015 which is underestimated in the model. The subsequent
stronger pulses, in November 2015 and March 2016, are reproduced but their magnitude are also slightly underestimated. In
general, the model captures the MBIs but has a tendency to underestimate salinity at the bottom, and overestimate it in the rest
of the water column (up to 35 m depth).
A time series of bottom salinity at the Arkona Buoy is shown in Fig. 10 (panel a), accompanied by observations at BY5
(Bornholm basin; panel b), BY10 (Gotland basin; panel c), and BY15 (Gotland deep; panel d). As stated above, the inflow
reaches Arkona on December 16. First observation of elevated salinity at BY5 is from February 19 2015 but due to the gap in
the measurements the pulse may have arrived earlier. At BY10 and BY15, an elevated bottom salinity is observed on February
21 and March 17, respectively. The timing is quite well captured by the model: At Arkona the salt pulse is delayed by 1.5
days. The modeled salt pulse reaches BY5 at the end of December 2014, BY10 in the end of February 2015 and BY15 in the
beginning of March 2015. The model underestimates the maximum bottom salinity by roughly 2.5 g/kg at Arkona, 0.5 g/kg
at BY5, 0.3 g/kg at BY10. The deviation, however, increases in time at BY5, BY10 and BY15. At BY5, the observed salinity



**Figure 9.** Vertical profile of salinity at Arkona. a) Observations; b) model; c) Difference (model - observations). The difference field has been computed by interpolating the model data to the observation locations.

remains above 18 g/kg (most of the time) after March 2015 while the modeled salinity decreases over time; in September 2016
the model underestimates salinity by 4 g/kg.
A similar decrease in modeled salinity, although smaller in magnitude, is also seen at BY10 and BY15. The observations
show a subsequent increase of bottom salinity (in February and April 2016 at BY15 and BY10, respectively) but the model
does not capture these events. In summary, the results indicate that the model does reproduce magnitude and timings of the
2014 MBI event and its propagation into the Baltic Proper. It does, however, underestimate the bottom salinity by 1 to 4 g/kg,
and does not reproduce lesser MBI events observed in 2015 and 2016. The negative salinity bias is also seen in the profile
statistics (Figure 8f).





**Figure 10.** Time series of bottom salinity at selected locations. The red line denotes the model, the black dots are the observations.



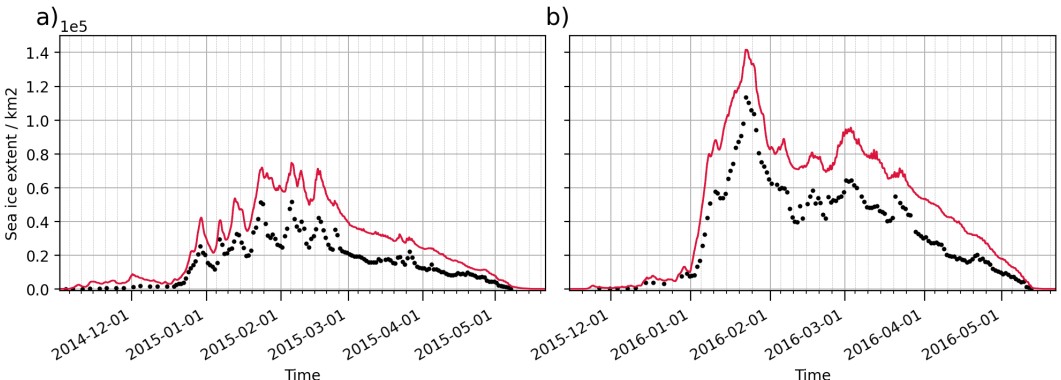

**Figure 11.** Time series of sea ice extent in the Baltic Sea for the two simulated winters. The red line denotes the model, the black dots are the observations.

## 3.5 Sea Ice

Sea ice extent (defined as the total area where sea ice concentration exceeds 15%) is presented in Fig. 11 for the winters 2014/2015 (panel a) and 2015/2016 (b). The winter 2014/2015 was exceptionally mild with only about 50 000 km$^2$ sea ice extent whereas the winter 2015/2016 was quite typical; maximum ice extent, 120 000 km$^2$, was observed in January 2016. During both winters, the model tended to overestimate the ice extent by roughly 25 000 km$^2$. In relative numbers, the maximum ice extent was overestimated by 45% and 25% for the two winters, respectively. The ice season also started earlier in the model, especially in November 2014.

## 3.6 SSH under storm conditions

To assess how well the model is able to reproduce sea level variations under storm conditions, we analyze the Elon and Felix storms that occurred between January 7 and 12, 2015 (Fig. 12). The storms created strong westerly winds in the southern Baltic, with daily mean wind speed between 10 and 18 m s$^{-1}$.

On January 8, westerly winds pushed water from the southern Baltic Sea to the east, lowering sea level in the Arkona basin and increasing it in the Gulf of Finland (event 1 in Fig. 12b,c); as the winds calmed, sea level at Helsinki retracted. The main storm event occurred on January 10, when strong westerly winds pushed water from the North Sea to Kattegat (Fig. 12d). Initially, sea levels rose in the Arkona basin (event 2A in Fig. 12c) but as winds prevailed and moved to the east, sea levels dropped by roughly 1.5 m in 12 h (2B). This led to a sea level increase in the Gulf of Finland and the Bothnian Sea (2B in Fig. 12a,b). On January 11, northerly winds pushed water to the south, causing an opposite sea level change (event 3). This was followed by another weaker westerly wind event (event 4).

Overall the model captured the wind-driven water elevation variations in the southern Baltic Sea. The extremes are slightly underestimated: the maximum observed and simulated SSH range at Rodvig are 1.68 and 1.55 m, respectively. At Ringhals the

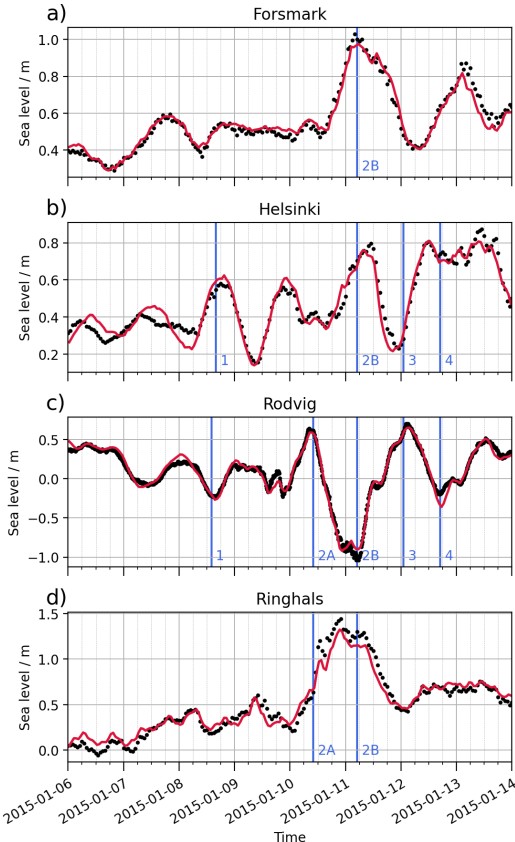

**Figure 12.** Time series of sea levels at selected locations during Elon and Felix storms. The red line denotes the model, the black dots are the observations. The model data has been bias corrected for visual comparison.

values are 1.40 and 1.27 m. The modeled seiche oscillations in Gulf of Bothnia and Gulf of Finland (Fig. 12a,b) are in good
agreement with the observations. In Helsinki, the amplitude tends to be slightly overestimated and there's a phase lag of 1 to
2 h. The tendency to underestimate SSH variability in the Danish Straits region, and overestimate it in the Baltic basin agrees
well with the results of long-term validation using tide gauge data (Fig. 3).
To assess how well the model captures high and low SSH events during the entire simulation period, we identified five
highest and five lowest SSH extremes in each tide gauge time series. The corresponding maximum/minimum SSH levels
were identified from the model within a 6 h window from the observation extremum. Model's deviation from the observed
peak/through was then calculated. Prior to the analysis, mean SSH was removed from both the observation and model data.
The results are summarized in Fig. 13.
The model tends to underestimate high SSH values especially in the tidally dominated regions (Danish Straits, Kattegat, and
Skagerrak). Low SSH values are similarly underestimated (i.e. positive deviation) in these regions. In general, the spread of the
deviation is larger (up to 70 cm) in the tidally dominated regions while being small in the deeper basins (within 20 cm). The



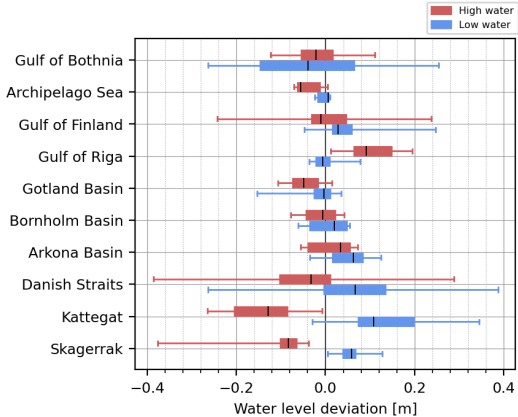

**Figure 13.** Model deviation in extreme water level cases in different sub-basins. Red and blue bars indicate model's deviation from the observed SSH maximum and minimum, respectively. Positive values mean that the modeled SSH is higher. The thin line denotes the entire deviation range; thick bars indicate the 25 to 75 percentile range. Black line denotes the mean. The sub-basins are defined as in Fig. 2.

spread is larger in the Gulf of Bothia and Gulf of Finland due to SSH build up in these elongated basins. High SSH events are overestimated in the Gulf of Riga. These results confirm that the model tends to underestimate SSH variability in the Danish waters while it is slightly overestimated in the Gulf of Bothia and Gulf of Finland.

## 4   Discussion and conclusions

The presented model validation for the 23 month period (November 1, 2014 – September 30, 2016) shows that the model reproduces SSH variability well. Model skill in the Baltic basin is especially good: CRMSD is typically below 7 cm. In the Danish Straits region, the deviation is larger due to the interaction of the tides and the complex topography. It is worth noting that many tide gauges are located in harbor or river mouth regions, or in small coastal embayments which cannot be properly resolved at the used resolution. The model has a tendency to underestimate SSH extremes especially under storm conditions. This is attributable to the atmospheric forcing data which generally tends to underestimate extremes during storm events, as well as to the model's dissipation (e.g., caused by coarse grid resolution, or friction parameterization).

The SST and SSS skill is generally good. The FerryBox comparison resulted in RMSD of 0.9°C and 0.70 g/kg, respectively. SST, however, shows a systematic negative bias of approximately -0.5°C. Our calibration runs suggest that the negative SST bias can be related to underestimated vertical mixing. Another possible cause is river runoff temperature which has a strong impact on SST in the vicinity of the river mouth. Yet another possibility is the downward longwave radiation forcing from the atmospheric model. The temperature deviation is larger in the bottom where RMSD can exceed 2°C.

The model tends to overestimate sea ice extent by 25 to 45%. As ice growth is strongly affected by SST, the overestimation is likely affected by the negative SST bias. Our calibration runs indicate that the sea ice model parameters have only a minor effect on the sea ice extent. Further research is needed to improve the biases in SST and sea ice.



The model replicates the 2014 MBI event and subsequent inflows in 2015 and 2016. The timing and magnitude of the MBI
salt pulse at Arkona compares well with observations and the bottom salinity is only slightly underestimated. In the Bornholm
Basin, the MBI is relatively well simulated by the model, especially considering that the initial bottom salinity is slightly low.
After the inflow, however, the modelled bottom salinity decreases at a semi-constant rate, and much faster than in reality. There
could be several reasons for this. First, it is possible that the vertical resolution is too coarse in the deeper layers (see discussion
below), leading to too high vertical numerical diffusion. Second, the fast decline in bottom salinity could be due to an absence
of smaller inflows after the MBI, which the model is not able to simulate well enough. Third, the type of vertical discretization
employed in the model ($z^*$ coordinates) is not so well suited to simulate dense bottom currents over rough topography, which
may lead to a spurious vertical circulation (Dietze et al., 2014). Smoothing of the topography between the Arkona Basin and
the Bornholm Basin may help in this respect.
The volume and propagation of the inflow can also be affected by the horizontal resolution: Inflow through the Sound, for
example, is significant, contributing 20 to 30% of the total volume (Mohrholz et al., 2015), yet it is difficult to resolve the
narrow strait in operational models (Fischer and Matthäus, 1996; She et al., 2007; Gräwe et al., 2015b).
While both the Nemo-Nordic 1.0 and 2.0 versions use 56 vertical levels, their thickness distributions are different. In the 2.0
configuration, the surface resolution has been increased from 3 to 1 m in the top 10 m, implying coarser resolution between 50
and 200 m depths. Our calibration runs with 75 and 100 vertical levels indicate that higher vertical resolution in 40 to 100 m
range improves bottom salinity in the Bornholm basin (not shown). In addition, the strength of the salt inflow is sensitive to the
bathymetry in the Danish Straits region (smoothing the bathymetry and deepening the channels), the used advection schemes
(UBS for momentum advection), as well as turbulence closure parameters (lowering the Galperin limit). Thus, horizontal and
vertical grid resolution, numerical mixing and turbulence parameterizations all play important roles in simulating the inflow
dynamics. MBI dynamics will be studied in more detail in the future.
From operational modeling perspective, the presented model configuration provides adequate skill for forecasting. SSH skill
is good in the entire Baltic basin and generally comparable to other models. It is worth noting that in short-term forecasts the
SSH skill is also highly dependent on the quality of the atmospheric forcing. SST and salinity biases are small or moderate and
those can be corrected with data-assimilation. Surface currents are an important product of operational forecasts, used in several
applications, such as oil drift modeling. Validating modeled currents, however, is challenging due to the lack of measurements
with sufficient spatial and temporal coverage. Comparing simulated currents at 1 nautical mile grid resolution against point
measurements poses additional challenges (Lagemaa et al., 2010). Nevertheless, the model's ability to simulate SSH, water
temperature and salinity demonstrates that the general circulation and dynamics are captured fairly well even though direct
validation of surface currents was not possible.
Improving the model skill is an ongoing effort. The bathymetry can be further improved to better represent the coast line
and shallow coastal regions, as well as unresolved channels. The wetting and drying capability of NEMO (O'Dea et al., 2020)
could improve SSH in shallow regions. Using higher vertical resolution will likely result in improved salt inflow dynamics.
Further work is needed to calibrate advection schemes, diffusion parameterizations, and the representations of overflows to
reduce numerical mixing. Most notably, adopting high resolution nesting in strategic regions, such as the Danish Straits and



the Archipelago Sea could greatly improve the representation of water exchange processes. Finally, data assimilation and online
coupling with atmospheric models can be used to improve both the forecast and reanalysis products.
*Code availability.* Nemo-Nordic is based on the NEMO source code version 4.0 (subversion trunk revision 11281), released under the open
source CeCill license (https://cecill.info, last access: March 29, 2021). The standard NEMO source code can be downloaded from the NEMO
website (http://www.nemo-ocean.eu/, last access: March 29, 2021). The Nemo-Nordic source code used in the present article has been
archived on Zenodo (Nemo-Nordic development team, 2021).
*Author contributions.* All authors contributed to the conceptualization of the model configuration, parameter tuning and validation. SF, PaL,
TK, LA, and IR implemented changes in the model source code, implemented the model configuration and processed forcing data. TK, PaL,
SF, LA, IR, IM, AL, SJS, SV contributed to model validation and tuning. TK was responsible for the original draft of the manuscript and
data visualization. TK, LA, IR, LT, VH, SF contributed to the review and editing of the manuscript. AN, VH, LT, PrL, and IL contributed to
the administration and supervision of the operational model development activities within the CMEMS BAL MFC project.
*Competing interests.* The authors declare that they have no conflict of interest.
*Acknowledgements.* This work is supported by the Copernicus Marine Environment Monitoring Service (CMEMS). Observational data was
provided by DMI, BSH, SHMI, MSI, LEGMA and FMI. This data was collected and made freely available by the Copernicus project and
programmes that contribute to it. Data analysis and visualization were carried out with Matplotlib (Hunter, 2007) and Iris (Met Office, 2010
- 2020) Python packages. The authors wish to acknowledge CSC – IT Center for Science, Finland, for computational resources.





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
