# Peer review of "Nemo-Nordic 2.0: Operational marine forecast model for the Baltic Sea"

_Geoscientific Model Development, 2021_

## Author Comment (AC1)

**Response to Reviewer 1**

Tuomas Kärnä et al.

August 18, 2021

In the following, all the comments raised by the Reviewer (in blue font) are addressed (in normal font). In the revised manuscript, all changes have been marked with red font color.

*Summary:*

*The manuscript describes an operational forecast model for the Baltic Sea and illustrates the applicability of the model by comparing a hindcast simulation to observational data from various sources.*

*Major comments:*

*The manuscript is well structured and the model seems well suited for the task at hand. I have only some minor points.*

*(1) The purpose of the model could be stated more clearly - which variables are of interest for the end-users and forecasted? In this context I would find it nice to motivate the model assessment metrics and the choice of the considered observations based on the model purpose (why are these of interest and which precision of the forecast is required for potential end-users?)*

We have revised the manuscript in this regard. The focus is now not only on forecasting but the development of a Baltic Sea model in general. The variables of interest are now mentioned in the abstract.

*(2) The assessment of the simulated sea ice is limited to a time series of sea ice extent. Some more assessment (including a visual impression of the spatial distribution) would be nice because I regard this aspect as very important for potential end-users.*

We agree. We have added a figure comparing sea ice concentration (Fig. 12 in the revised manuscript).

*More specific comments are listed below. Note that I am not an English native speaker either.*

*Specific Comments:*

*Line 1: The authors could add one sentence which model output they want to provide to the end user.*

Added.

*Line 7: I would suggest to add that the comparison is based on a hindcast simulation and the considered time period.*

We have added the time period in the sentence:

*The model is validated against sea level, water temperature, and salinity observations, as well as Baltic Sea ice chart data for a two-year hindcast simulation (October 2014 to September 2016).*

*Line 17: Please add Dietze et al. (2014) (already referenced later)*

Added.

*Line 50ff: For my feeling the part about spurious mixing is a bit long in the introduction and parts of it could be moved to the final discussion when referring to the representation of inflows from the North Sea.*

We have revised the introduction and shortened the discussion of numerical mixing. It is an important topic, however, and often overlooked by model users, and thus we feel that it is worth mentioning in the introduction.

*Line 64ff: The following parts seems a bit unrelated to the foregoing text. Maybe the authors could add a few sentences why operational forecasting is required in the Baltic and then come to the operational models.*

We have revised the introduction for clarity.

*Line 70: better ". . . .ocean circulation model. . . ."*

We have added the word ocean in the sentence.

*Line 80ff: I would suggest to add what should be forecasted/is of interest to potential end users.*

We have formulated the paragraph to emphasize the model configuration and its generic use in both hindcast and forecast tasks. This is now mentioned in the abstract as well. The fields of interest are mentioned in the subsequent paragraph, so we are not duplicating those here:

*The aim of this article is to validate the Nemo-Nordic 2.0 model configuration. The configuration is used in the EU Copernicus Marine Service for both near-realtime forecasts, as well as multi-year hindcast simulations for the Baltic Sea. The presented validation is based on a 2-year hindcast simulation that uses similar forcing as the operational configuration. [...]*

*Line 86: Instead of "skill metrics" I would rather use "model assessment metrics" - especially since the model is evaluated on a hindcast simulation and no real forecasts are considered. Note that this expression occurs rather often in the text and should be changed consistently.*

We have changed the terminology as suggested (also elsewhere in the paper). We do, however, continue to use the term "model skill" where appropriate: The term skill is not restricted to solely the forecast task.

*Line 90: I find this confusing. Better? "..is an updated version of Nemo-Nordic 1.0 based on..."*

The sentence has been revised:

*The presented model setup is an updated version of the Nemo-Nordic 1.0 configuration (Hordoir et al., 2019) implemented on NEMO version 4.0 (Madec et al. 2019; subversion repository revision 11281).*

*Line 93: horizontal resolution*

Fixed.

*Line 93ff: I would finish the description of the model grid before coming to the boundaries. Also, it should be specified which open boundaries are used.*

The location of open boundaries is a property of the horizontal grid being presented. We have thus not altered the text. The open boundary types are now mentioned in Section 2.3.

*Line 99: was -> were*

Fixed.

*Line 102: How was the bathymetry modified (what counts as "shallow")?*

We have revised the sentence:

*The bathymetry was modified along the west coast of Denmark by masking out shallow lagoons and channels (such as the Wadden Sea, Ringkøbing Fjord and Limfjord area; cut-off depth was about 10 m; Fig 1 c) to improve the propagation of tides to Skagerrak and Kattegat.*

*Line 106: How was the tuning done?*

The tuning process is explained in the same section. We have now revised the text for clarity: description of bottom friction tuning now follows the sentence:

*The model configuration was tuned to accurately simulate surface gravity waves and internal gravitational currents. Bottom friction is [...]*

*Line 135ff: I would suggest to finalize the description of the ocean part before describing the sea ice.*

Good idea, the order has been updated.

*Line 173: I would rather call this section "Boundary conditions"*

The section is now called "Boundary conditions and forcings"

*Line 174ff: The initial conditions could be described in a bit in more detail (e.g. from which conditions did the spin-up start?) because I expect at least some impact on the representation of the simulated deep water properties. Please note in this context that I see no perfect solution for their choice because the Baltic Sea virtually never reaches some kind of steady-state.*

We have elaborated the description of the spin-up run in Section 2.3.

*Line 178: Better? "The atmospheric boundary conditions are provided by ...". Also, I would be interested in the spatial resolution of the atmospheric forcing.*

We prefer the more active voice wording ("model is forced by" rather than "conditions are provided by"). We have added the resolution of the atmospheric model (3 km) to the text.

*Line 182/183: Better move to line 180 (+ delete "in NEMO")*

Text has been revised.

*Line 190ff: Better? Observational data were provided by ....*

Also here we prefer "obtained from" rather than "provided by".

*Line 208: Better? "Model assessment metrics"*

Changed.

*Line 231: I am not sure what is meant by "datum" in this context.*

Vertical datum is the reference zero level in digital elevation models. We believe it is standard terminology.

*Line 235ff: The following part sounds like a lengthy excuse why SSH might not be captured perfectly in the model - which for me would rather refer to the discussion of the model results. Also, I would suggest to express this in a more positive way, i.e. what to expect from a SSH forecast. For my feeling the most important aspect for end users are deviations from the mean.*

We have shortened this part of the text:

*The exact vertical reference datum of the circulation model is not well defined. Consequently, SSH bias cannot be reliably evaluated and we therefore assess SSH performance with centralized metrics, i.e. CRMSD and Taylor diagrams.*

*Line 252: When mentioning the locations (Kiel Holtenau etc.) the authors could refer to Fig.1c.*

We have added a reference to Fig 1 in the captions of Figs. 2 – 5.

*Line 259: This reads a bit confusing for me. Suggestion: The agreement between model and the ssh observations is generally higher in the open Baltic Sea than in Danish Straits. In the open Baltic the NCRMSD is generally below 0.3 and correlations between model and observations are above 0.95 (exceptions are....). In the Danish Straits, stations Fredericia and Copenhagen show much lower correlations of ...., respectively. This local drop in the correlations is expected due to the complex bathymetry in the Danish Straits.*

We have revised the text as suggested.

*Line 263ff: The BSH provides 2D maps for SST. It might be nice to show an example snapshot – even though the statistical value is of course limited.*

The model is running operationally at CMEMS so examples of SST fields can be obtained from their services. As the Reviewer states, including a map of SST in the paper without a comparison or analysis does not seem, in our opinion, to add much value.

*Line 265ff: Maybe add "..mixing in the water column" because the other processes mentioned here refer to the atmosphere.*

Added.

*Line 265/266: ?*

It is unclear what the Reviewer is referring to by the question mark. Nevertheless, we have revised the wording in this paragraph.

*Line 281: The authors could add that they refer now to one of the ferries (for those who are not so familiar with ferry box data and the respective shipping lines). At which depths do the ferries measure?*

Ferries are mentioned already in the previous paragraph. TransPaper measures at 5 m depth, FinnMaid at 3 m. This information has been added to Section 2.4.

*Line 288/289: Which gradient?*

Horizontal, added.

*Line 294ff: I would expect that the deep water properties are still impacted by the initial conditions because Döös et al. (2007) report a residence times of almost 30 years for the Baltic Sea. This can in my eyes not be avoided here but could be mentioned (unless this issue was investigated somehow and ruled out).*

We agree. We mention this now in the discussion section.

*In this section I personally would include an assessment of the simulated mixed layer depths.*

See our reply to Reviewer 2 on the same topic: We agree that comparing the thermo- and halocline structure is surely valuable, but doing it reliably is out of the scope of the present paper.

*Line 299: I would use some other wording instead of "Temperature profile skill" (similar expressions are used several times.)*

Replaced by "temperature profile metrics", also elsewhere.

*Line 357ff: I regard the representation of sea ice as very important in a forecast model. I would find it nice to see a/some 2-dimensional map(s).*

We have added maps of sea ice concentration.

*Line 392ff: For my feeling the discussion could focus more on the applicability of the model for forecasts. I like the honest discussion about MBIs.*

The paper is now revised to not only focus on forecasting. Thus the discussion addresses model performance in general.

*Line 444: Why would a forecast model require a coupling with an atmospheric model? A coupled ocean-atmosphere model is not suitable for a 1:1 model-data comparison.*

We have changed the focus of the text: the model is not solely designed for the forecast task.

*Line 445: Code availability: Pease provide doi and the link. I would find it nice if also the extra information to run the model was provided in addition to the code (such as initial conditions (restart), short test forcing/boundary conditions and specific model settings).*

The DOI and the link to zenodo.org are provided in the citation. The model input files are included in the code archive.

We fully agree that including initial conditions and forcings etc. would be very beneficial to the community. However, unfortunately due to licensing issues we do not have the rights to redistribute the data.

*Reference*

*Döös, K., & Engqvist, A. (2007). Assessment of water exchange between a discharge region and the open sea–a comparison of different methodological concepts. Estuarine, Coastal and Shelf Science, 74(4), 709-721.*

---

## Author Comment (AC2)

**Response to Reviewer 2**

Tuomas Kärnä et al.

August 18, 2021

In the following, all the comments raised by the Reviewer (in blue font) are addressed (in normal font). In the revised manuscript, all changes have been marked with red font color.

*Revision of Nemo-Nordic 2.0 by Kärnä et al.*

*Summary:*

*The authors present and evaluate Nemo-Nordic 2.0, an operational marine forecast model of the Baltic Sea, which is based on Nemo-Nordic 1.0. The evaluation is sound and covers the most important aspects of the Baltic Sea physical oceanography, and the paper has overall a good structure and is in general well written. This is an important paper that documents the development of the Nemo-Nordic configuration, and deserves being published after revision.*

*Major comments:*

*-The introduction is not very coherent and needs to be revised. Maybe you could divide it into two subsections, with one about the modelling, and one more detailed about the physical oceanography of the Baltic and the North Sea? Alternatively, you could shorten the part on physical oceanography and only keep the most important aspects. At the moment there is a lot of information on the physical oceanography in there without references (lines 25-34), please add appropriate references that support your description if you want to keep this text.*

We have revised the introduction. It is now shorter, focusing on numerical modeling with less discussion about the physical oceanography of the system.

*-The model covers the Baltic Sea and the North Sea, but you only evaluate its performance in the Baltic Sea, and thus only a part of the model. Please describe why you do this. Still, I think that it is important, and that would be of value, if you also evaluate the model in the North Sea. If you want to focus on the Baltic Sea only in the main manuscript, you could maybe put some figures in supplementary information?*

The model has been developed in the CMEMS BAL MFC project, where the goal is to model the Baltic Sea. As stated in the introduction, modeling the Baltic Sea requires a large portion of the North Sea as well due to the tight coupling of these seas. This is especially important for the modeling of tides and weather-induced water elevation variability across the Danish Straits region. We have added a comparison against 4 tide gauges in the North Sea to the SSH Taylor diagram (Fig. 3). The results show that the tides in the North Sea are captured with similar accuracy as in Skagerrak and Kattegat.

We agree that the Nemo-Nordic 2.0 model could potentially be used for operational purposes in the North Sea as well, and should therefore be thoroughly evaluated there as well. This is a laborious task, however, and out of the scope of the present paper.

*-You write that in this paper you evaluate the Nemo-Nordic forecast system. But, doesn't a forecast system also include data assimilation and forecasts? (or hindcasts). Indeed, you call your simulation a hindcast, but you do not evaluate its ability to "predict" the past, i.e. for how long the model manage to reproduce observations if starting from initial conditions created with data assimilation. I do not think that you need to do this in the paper, and that it would be a paper of its own. I think that this is just a question of adding some extra text in the discussion/introduction about this, and/or revising the choice of words.*

We have revised the manuscript in this regard: the focus is now on Baltic Sea modeling in general, not only on the forecast task.

*-Wouldn't it be interesting to show how Nemo-Nordic 2.0 performs in comparison to Nemo-Nordic 1.0?*

It would. This is, however, not straightforward as the Nemo-Nordic 2.0 configuration is a completely new setup implemented on top of the NEMO 4.0 version. Thus, we do not have comparable model configurations: The previous Nemo-Nordic 1.0 runs used different grids, forcings, time periods, and had different model parameters. In order to carry out a reliable comparison, we should essentially port the presented 2.0 configuration (bathymetry, forcings, and model parameters) back to NEMO 3.6 which implies a substantial amount of work. Despite its usefulness, we therefore chose not to pursue such a comparison.

*Minor comments:*

*lines 4-5: the 1 nautical mile horizontal resolution is an update as well no?*

Hordoir et al. (2019) present both 1 and 2 nautical mile versions of Nemo-Nordic 1.0.

*lines 15-17: These models do not only simulate the circulation... maybe it is better to write: "several ocean circulation models have been set up for the Baltic Sea", or something similar*

We have reformulated the sentence as suggested.

*lines 84-85: you repeat "as well as" twice in one sentence, please revise*

The text has been revised.

*In section 2 it is not very clear what settings that are updates since Nemo-Nordic 1.0, and what you have kept the same, please clarify this.*

We have added a paragraph summarizing the main differences to the beginning of Section 2.1.

*line 110: do you resolve baroclinic eddies in your configuration?*

Only in the larger basins, where the internal Rossby radius of deformation is approximately 3 to 7 km. Our 1.8 km grid can resolve the larger eddies in this range. Nevertheless, the numerical schemes that decrease numerical mixing in the eddying regime are likely to perform well in the case of unresolved eddies as well.

*line 174: 14 months are not enough to spin up the deep Baltic Sea. What did you use to initialize the spin-up run?*

The description of the spin-up run is described in more detail in Section 2.3.

*line 244, and also elsewhere in the manuscript: how do you define good, relatively good and other, similar, qualitative words?*

That is a good question. The notion of "good" skill in the manuscript is based on a) the previous modeling expertise of the authors, and b) the skill metrics. We do not present quantitative comparisons against previous modeling efforts, because the modeled time periods and model configurations are different and hence a direct comparison is not meaningful. In terms of the metrics, the normalized Taylor diagram gives a good overview of the statistics of the (centered) deviation. Also, as stated in Section 2.5, NRMSD is a particularly useful metric for defining "bad" skill in general terms (i.e. NRMSD¿1). But as the complex model behaviour can never be expressed with a single metric, we consider several of them simultaneously (e.g. in figs. 2-5 and 8).

*figure captions and in the text discussing these figures: please describe if it is based on daily or monthly output. It should also be mentioned in the methods what time-frequency your analyses are based on. A bit in a similar manner as you do for the comparison with the Ferrybox data.*

The model output was stored at 1 h resolution (now mentioned in Section 2.1). The tide gauge data was used in its original temporal resolution, i.e. 10 min, 15 min, or 1 h (added to Section 2.4). Prior to computing the error metrics, the model outputs were interpolated to the observation time stamps (as

already mentioned in Section 2.4).

*Could you put some lines in figure 1 showing the routes of TransPaper and Finnmaid ferries? As it is now it is difficult to relate figures 6-7 to a geographic location.*

Added.

*I like figure 8, but I have some questions related to it; Why did you choose the upper 10 m for the surface layer? The summer thermocline is generally located deeper. When showing the skill as you do for the surface and deep layer, it does not tell us if the bias is due an eventual mis-placement of the thermocline/halocline, or if it is the modelled temperature/salinity that is off. It would be valuable if you could evolve the text around this and discuss it, a bit like you do for the salinity at BY5.*

We agree that comparing the thermo- and halocline structure between the model and observations is very valuable. However, this is a complicated task as the criteria for detecting the clines depend on the region (e.g. the Bothnian Bay and the Gotland Deep), and in many cases the water column is stratified with several possible thermo/halocline locations. The assessment, then, is highly dependent on the chosen metrics. We agree that such an analysis would be of interest, but it should be done separately for each sub-basin. This is a topic for future publication.

In the present paper, we have taken a simpler approach: The RMSD of the whole column indicates general agreement, and the top/bottom bins aim to assess the similarity of the surface/bottom waters.

*figure 9 and 10: please write that the difference shows the model-observations*

This was already mentioned in the caption of Figure 9. Now it reads: *[...] c) Difference (model minus observations).* Figure 10 does not have a difference signal.

*section 3.6: please write why you have chosen these specific locations for your analysis*

We have added a sentence:

*Figure 13 shows SSH time series at four stations in different parts of the Baltic Sea to illustrate the propagation of seiche oscillations.*

*figure 12: describe in the caption what the blue lines show.*

We've added a sentence in the caption:

*The blue vertical lines indicate events discussed in the text.*

*from figure 9 it looks like the model is too diffuse in the vertical. Maybe you could add this to your discussion on lines 408-416.*

We have emphasized the role of vertical mixing in the discussion.

*lines 429-430: do you have some references to other baltic sea operational models that you could put here?*

We have added references to other models in this paragraph:

*From an operational modeling perspective, the presented model configuration delivers sufficient performance, generally comparable to other models (e.g., Meier et al., 1999; Burchard et al., 2009; Dietze et al., 2014; Hordoir et al., 2019).*